# Depth-Based Intervention Detection in the Neonatal Intensive Care Unit Using Vision Transformers

**DOI:** 10.3390/s24237753

**Published:** 2024-12-04

**Authors:** Zein Hajj-Ali, Yasmina Souley Dosso, Kim Greenwood, JoAnn Harrold, James R. Green

**Affiliations:** 1Systems and Computer Engineering, Carleton University, Ottawa, ON K1S 5B6, Canada; yasminasouleydosso@cmail.carleton.ca (Y.S.D.); jrgreen@sce.carleton.ca (J.R.G.); 2Clinical Engineering, Children’s Hospital of Eastern Ontario, Ottawa, ON K1H 8L1, Canada; greenwood@cheo.on.ca; 3Neonatology, Children’s Hospital of Eastern Ontario, Ottawa, ON K1H 8L1, Canada; jharrold@cheo.on.ca

**Keywords:** depth camera, neonatal patient monitoring, NICU, transformer, vision transformer, ViT, intervention detection

## Abstract

Depth cameras can provide an effective, noncontact, and privacy-preserving means to monitor patients in the Neonatal Intensive Care Unit (NICU). Clinical interventions and routine care events can disrupt video-based patient monitoring. Automatically detecting these periods can decrease the time required for hand-annotating recordings, which is needed for system development. Moreover, the automatic detection can be used in the future for real-time or retrospective intervention event classification. An intervention detection method based solely on depth data was developed using a vision transformer (ViT) model utilizing real-world data from patients in the NICU. Multiple design parameters were investigated, including encoding of depth data and perspective transform to account for nonoptimal camera placement. The best-performing model utilized ∼85 M trainable parameters, leveraged both perspective transform and HHA (Horizontal disparity, Height above ground, and Angle with gravity) encoding, and achieved a sensitivity of 85.6%, a precision of 89.8%, and an F1-Score of 87.6%.

## 1. Introduction

The Neonatal Intensive Care Unit (NICU) provides critical care for the most vulnerable newborn patients. Such patients are characterized by precarious health and require continuous monitoring. Such continuous monitoring in the NICU typically involves sensors attached to the patient’s skin, which are susceptible to motion artifacts and may interfere with both clinical and parental care. The wired sensors can irritate sensitive skin, with frequent removal and reapplication sometimes required during medical interventions. This motivates the development of robust video-based noncontact patient monitoring [1,2,3].

A patient may experience multiple periods of clinical intervention or routine care throughout their time in the NICU. These interventions can include a clinician or parent reaching into the scene to replace sensors, take readings, change a diaper, feed the patient, or otherwise move the patient. These periods of intervention are often excluded from analysis when studying novel noncontact techniques of monitoring neonates in the NICU (e.g., [4,5]). However, studies by Villarroel et al. [3] and Souley Dosso et al. [2,6] attempt to detect these periods of intervention and, in the case of [6], classify a subset of them (bottle-feeding interventions).

Deep learning has led to dramatic advancements in computer vision, which have translated into new forms of noncontact patient monitoring [1]. Souley Dosso et al. [2] used the VGG-16 CNN model introduced in ref. [7] as the feature extractor for their method of intervention detection. They examined several forms of multi-modal (RGB and depth) fusion, resulting in similar performance between the RGB and RGB-D fusion models while observing significantly lower performance for depth-only models.

In this paper, we develop a model to detect periods of clinical or routine care intervention using only depth-based images, as this modality is more privacy-preserving than RGB or RGB-D images. Detecting such interventions is useful for several reasons. For example, when estimating vital signs, estimation may be paused or patient monitor alarms may be silenced automatically during interventions since a clinician is already attending to the patient. Detecting interventions is a step towards classifying interventions, which may ultimately lead to automated charting of patient care. Furthermore, by creating an intervention detection system based strictly on depth data, detection will be robust to changes in lighting, which can occur frequently in the NICU due to clinical interventions and parent time and regularly throughout the day for some premature patients to promote sleep and support development. This paper focuses on utilizing depth video alone for intervention detection, building on the preliminary results reported in the following thesis [8]. Note that portions of this manuscript previously appeared in the following thesis [9].

This study leverages vision transformers (ViTs) [10], which have been shown to outperform CNNs for image classification in several application areas. The ViT divides each input image into a number of nonoverlapping patches which are flattened into vectors of pixel values and used as the input to the transformer’s encoder. The ViT culminates in a fully connected head layer for the task of image classification. Variations and extensions of this model have had success in image segmentation, object detection, and video action recognition [11,12,13].

When training a deep learning model, large amounts of data and compute resources are needed. For this reason, transfer learning is usually employed, where models are pre-trained on large datasets prior to fine-tuning the model to perform specific tasks with smaller training datasets. For image classification, several convolutional neural network (CNN) and vision transformer (ViT) models are available that have been pre-trained on the ImageNet dataset [14] consisting of ∼14 million annotated images from 1000 classes. Given a downstream task, pre-trained models are normally chosen from the same or similar domains (e.g., RGB image classification, object detection, semantic segmentation). Transfer learning has been shown to improve the average accuracy of CNN models [15] as well as ViT [10] for image classification.

Pre-trained image classification models are generally trained on large amounts of labelled three-channel RGB data. HHA encoding is a method of encoding depth data using three channels for each pixel rather than just the one channel of depth [16]. An example illustrating the three channels resulting from HHA encoding of a depth image can be seen in Figure 1. The three channels correspond to the horizontal disparity (H), the height above the ground (H), and the angle the pixel’s local surface normal makes with the inferred gravity direction (A). This has been shown to improve the performance of a network pre-trained on RGB data and fine-tuned with labelled HHA-encoded depth data when compared to fine-tuning on regular one-channel depth or disparity data. Gupta et al. suggest that this is because the disparity and angle channels may show edges that correspond to object boundaries that can be seen in the RGB images of the same scene [16]. The authors verify this by fine-tuning a CNN originally trained for object detection and semantic segmentation from RGB images [17].

The horizontal disparity can be calculated from depth data by using Equation (Equation 1) [18]:(1)$$\mathrm{Disparity}=\frac{\mathrm{Focal}\mathrm{Length}\times \mathrm{Baseline}}{\mathrm{Depth}}$$
where the *Focal Length* and *Baseline* are found from the camera’s intrinsic matrix [19]. The height above the ground and the angle between the surface normal and inferred gravity direction can be found using the algorithms presented in ref. [20] and implemented in refs. [21,22]. The algorithms require the point cloud representation of the depth image as well as the camera matrix. The direction of gravity is estimated by finding the direction that is best aligned to surface normals, under the assumption that most surfaces in the scene are horizontal. The direction of gravity is initialized to the camera’s Y-axis before iteratively refining the estimated direction by examining local surface normals in the depth data. The height above the ground can then be found by rotating the point cloud of the data to the horizontal direction then subtracting the smallest Y-coordinate value in the scene [23]. The angle between the surface normal and the gravity direction can be found from the difference in the respective vectors. Finally, the values in each of the channels are normalized to the range of 0–255 (i.e., an 8-bit value).

Despite the notable advancements in noncontact monitoring of patients in the NICU, there remains a critical gap in the literature concerning the automatic detection of clinical interventions and routine care events, particularly using depth data. Many existing studies [2,3,6] have leveraged RGB (colour) or multi-modal RGB-D image data for such detection. Chaichulee et al. achieved excellent accuracy when detecting clinical interventions using RGB video [24]. However, RGB (and RGB-D) video may be considered intrusive and is sensitive to ambient lighting. Therefore, in this study, we specifically focus on models restricted to privacy-preserving depth images rather than RGB (or RGB-D).

The contributions of this study are as follows: First, we propose an intervention detection method based solely on depth data, thereby increasing robustness to lighting changes and maintaining patient privacy. The method utilizes a vision transformer (ViT) model to interpret the depth data, an approach not previously explored for this application in the NICU setting. We also investigate several design parameters such as the encoding of depth data and the application of perspective transform to account for varying camera placement, hence offering a versatile solution suitable for different NICU environments. Finally, we evaluate our model using real-world NICU data, demonstrating its practical utility and efficacy. Utilizing real-world data for the evaluation is critical in this case, as a simulated environment may not accurately reflect the range of challenges that arise in a complex clinical environment. An example of several challenging scenarios can be seen in Figure 2. Our results not only confirm the feasibility of the proposed approach but also set the stage for future work in automatic classification of interventions and eventually automated charting of patient care.

## 2. Materials and Methods

### 2.1. Data Collection

To support our study, we collected two types of data: clinical data from neonatal patients and simulated data from a neonatal manikin. In the following subsections, we describe the data collection process for each dataset, including details on the data collection setup, data processing, and class labelling.

#### 2.1.1. NICU Data Collection

Data were collected from 27 neonatal patients in the NICU at the Children’s Hospital of Eastern Ontario (CHEO) following approval by the research ethics boards from the hospital and Carleton University. The data were collected as part of a larger research initiative to develop multi-modal noncontact patient monitoring methods and technologies. The dataset cannot be released publicly due to the restrictions set by the research ethics board.

Figure 3 shows an example of the setup in the NICU environment. An RGB-D camera (Intel RealSense SR300, Santa Clara, CA, USA) was placed above or beside the patient’s bed. The camera was chosen due to its small size, affordability, and suitable depth range to capture patients at a close distance. Recordings were captured at a resolution of 640 × 480 pixels at 30 frames per second. The cameras were placed such that the view planes were at nonuniform angles relative to the plane of the bed. The SR300 captures depth information using the coded-light method; using a combination of an IR projector and IR camera sensor to generate a depth pixel frame. The camera also includes a separate RGB camera sensor that can be used in conjunction with the depth stream to form an RGB-D image. Note that in the present study, all proposed methods use only privacy-preserving depth image data.

The gold standard respiratory rate signals of the patients were recorded from the bedside patient monitor (Draegar Infinity Delta). Custom Patient Monitor Data Import (PMDI) software Version 1.0, developed for the project, was used to import the data from the serial port on the monitor [25]. A bedside annotation application was used to annotate events (clinical interventions, etc.) in real time. All data from the camera and patient monitor were saved on a data acquisition laptop.

Still images were extracted from the patient recordings every 30 s and labelled as either ‘Intervention’ (positive) or ‘No Intervention’ (negative). This resulted in 14,892 images in total, with 1260 in the positive class and 13,632 in the negative class (a class imbalance of 10.8:1 in favour of the negative class). The ‘Intervention’ class comprised images where a nurse or other practitioner was reaching into the camera’s view to tend to the patient, while the ‘No Intervention’ class included only the patient (Figure 4).

The difficulty of intervention detection from depth data can sometimes be misrepresented. Looking at Figure 4, one would assume that the difference in the depth frame between the nurse’s hands and the patient/bed would be apparent; however, the task is often more difficult. In Figure 5, an intervention frame can be seen that is more challenging to classify by looking only at the depth channel (on the right). If the caregiver’s hands are near or at the height of the patient’s bed, the difference in depth can be sufficiently small to require more advanced methods. This is demonstrated by including a baseline approach in the present study.

#### 2.1.2. Simulated Data Collection

After the initial data collection stage, additional simulated data were collected to partially address the class imbalance between nonintervention/intervention frames in the clinical data. A neonatal manikin (StandInBaby [26]) was placed on simulated clinical bedding, and the RealSense SR300 RGB-D camera was used to capture 600 depth images, as illustrated in Figure 6. A camera arm was used to place the camera at 5 different angles relative to the plane of the bed. Yellow gloves were worn during data collection to facilitate the use of the collected data in image segmentation studies in future studies by providing a consistent colour reference for the hands (Figure 7).

### 2.2. Proposed Method

Vision transformers have demonstrated excellent accuracy in image classification tasks since their introduction in ref. [10]. We propose the use of a ViT pre-trained on the ImageNet dataset [14] and fine-tuned on a subset of our own set of 14,892 depth images. The model architectures were implemented using the PyTorch Image Models library [27]. Two model sizes with similar architecture but different numbers of trainable parameters were chosen, ‘vit_tiny_patch_16_224’ and ‘vit_base_patch_16_224’, with ∼5.4 M parameters and ∼85 M parameters, respectively. Each of the models accepts input images with a resolution of 224 × 224 pixels and divides them into 16 × 16 patches for embedding. The difference in the number of trainable parameters comes from an increase in the dimensions of the hidden embedding layer and the number of heads in the attention mechanism when moving from the ‘tiny’ model to the ‘base’ model.

*Performance evaluation*: Throughout this study, we have used a repeated five-fold cross-validation approach, where the dataset of 27 patients was divided into five distinct “folds”. For each combination of system design parameters, five models were trained and evaluated, and the average performance across the five models was computed. Within each of the five folds, a classification model was trained on four folds (approximately 21 patients), while the remaining 5–6 patients *not used to train the model* were used to evaluate the model. In this way, all patient data were used to both train and evaluate models but never the same model. This entire process was repeated five times, with different patients assigned to each fold in each repetition. The mean across the five repetitions was reported as the final performance metric.

*Hyperparameters*: The training of the models utilized a mini-batch size of 16 and a learning rate set at 0.01 over a maximum of 15 epochs. Visual inspection of preliminary learning curves indicated no substantial reduction in validation loss beyond 15 epochs. Stochastic gradient descent was employed as the optimizer, with a momentum of 0.9. The input images were each resized to dimensions of 224 × 224 pixels, which altered their aspect ratio from 4:3 to 1:1. Finally, random rotations (ranging from 0° to 360°) and horizontal/vertical flips were applied to the images in the training sets.

Along with the size of the model, the effect of three other system design parameters on model performance was also explored. These parameters are described in the upcoming Section 2.2.1, Section 2.2.2 and Section 2.2.3, and a summary can be seen in Table 1. A visual representation of the data flow and utilization of the proposed system design parameters can be seen in Figure 8.

#### 2.2.1. Simulated Data

Since the data collected from the NICU contain more instances without interventions than those with, the resulting labelled data had a high class imbalance of 10.8:1 in favour of the negative (no-intervention) class. To help correct for this imbalance, simulated intervention data were collected as previously described (Section 2.1). These data comprised 600 images of simulated interventions that were added to the positive class, bringing the class imbalance down to approximately 7.3:1. Both model sizes were trained without the addition of the simulated data, and then the process was repeated with the inclusion of the simulated data in each training fold (i.e., simulated data were used for training but not for testing).

#### 2.2.2. Perspective Transformation

The effect of a perspective transformation (PT) algorithm on the performance of the models was explored. As discussed in ref. [5], we previously demonstrated that perspective transformation can account for nonoptimal depth camera placement relative to the patient. In that study, perspective transform was shown to improve an ROI selection algorithm for subsequent respiration rate estimation. Based on those results, it was thought that applying the transformation to the data used to build the ViT-based intervention detection model might also improve its performance. The patient data collected from the NICU and the simulated data were transformed by manually selecting four registration points in the plane of the bedding for each new recording. The rotation matrix was found and applied to all frames extracted from the same recording. The experiments were then re-run using these transformed data as the input. Models were trained and tested with and without perspective transform to investigate its effect on intervention detection accuracy.

#### 2.2.3. HHA Encoding

ViTs are not typically trained from scratch for specific image classification tasks. Rather, ViT models are typically pre-trained on large datasets using self-supervised learning techniques, such as masked auto-encoding (MAE) [28]. Pre-trained ViTs are then fine-tuned for specific tasks through the addition of a task-specific prediction head. Such pre-training of ViT requires a large amount of data and extensive compute resources. Some ViT models pre-trained on large image datasets, such as ImageNet, have been released publicly by researchers at Google Research [29] and other groups. As these models have been pre-trained on 3-channel RGB images, there is latitude as to how the single channel of depth data should be mapped to a 3-channel input. The effect of HHA encoding on the performance of the proposed intervention detection model was investigated.

Each of the datasets described previously was transformed to be HHA-encoded, and the experiments were re-run. Models were trained with and without HHA encoding to investigate its effect on intervention detection accuracy. Models trained without HHA encoding were modified to accept 1-channel images as inputs. The pre-trained input layer weights from each of the 3 channels normally used for R, G, and B were summed into a single channel.

### 2.3. Baseline Methods

The models explored in this study were compared against the best-performing CNN-based intervention detection model proposed by Souley Dosso et al. in ref. [2]. Specifically, the model chosen for comparison was the multi-modal RGB-D fusion model, which used a VGG-16 CNN architecture [7] and was pre-trained on the ImageNet dataset [14] and fine-tuned on the intervention detection dataset described in Section 2.1.

Additionally, the exclusively depth-based model from Souley Dosso’s study was included for comparison given its shared reliance on depth modality, though it resulted in lower performance metrics overall. For this model, the VGG-16 input layer was modified by removing two of its three input channels, allowing the pre-trained weights to be fine-tuned on the single depth channel. Further, a conventional (rules-based) method was also evaluated as an alternative baseline for comparison. The method consists of designating a known nonintervention frame for each patient recording and calculating the mean squared error of each of the rest of the frames.

As a final baseline model for comparison with our depth-based models, the RGB-D and depth-based models presented in ref. [2] were also re-trained and evaluated using the design parameters outlined in Section 2.2.1, Section 2.2.2 and Section 2.2.3. This enabled direct comparisons between the depth-based vision transformer models proposed here and Souley Dosso’s depth-based CNN models for each of the design variables explored in this study.

## 3. Results

Each of the models was evaluated using 5-fold cross validation repeated five times. Each fold contained data from unique patients, leaving data from five or six patients as the test set each time. The frames were extracted at the same time points in the videos as the data used in ref. [2] to enable direct performance comparisons against the chosen baseline models. For experiments where simulated data were used, the simulated frames were added to the training set in each fold. The metrics used to evaluate the models were specificity, sensitivity, precision, accuracy, F1-score, and Matthew’s correlation coefficient (MCC) (Equation 2)–(Equation 7). Analysis of Variance (ANOVA) was run on the results from the proposed models to determine the statistical significance of the effects for each of the design parameters. This was performed by collapsing the results of the repetitions of each fold by calculating the average of each metric before running the ANOVA test. This meant that the number of records was reduced from 400 (5 folds × 5 repetitions × 16 combinations of variables) down to 80 (averages of the repetitions of the 5 folds × 16 combinations of variables). The full ANOVA test results can be found in Appendix A.
(2)$$\mathrm{Specificity}=\frac{\mathrm{TN}}{\mathrm{TN}+\mathrm{FP}}$$
(3)$$\mathrm{Sensitivity}=\frac{\mathrm{TP}}{\mathrm{TP}+\mathrm{FN}}$$
(4)$$\mathrm{Precision}=\frac{\mathrm{TP}}{\mathrm{TP}+\mathrm{FP}}$$
(5)$$\mathrm{Accuracy}=\frac{\mathrm{TP}+\mathrm{TN}}{\mathrm{TP}+\mathrm{FP}+\mathrm{TN}+\mathrm{FN}}$$
(6)$$F1\text{-}\mathrm{Score}=\frac{2\mathrm{TP}}{2\mathrm{TP}+\mathrm{FP}+\mathrm{FN}}$$
(7)$$\mathrm{MCC}=\frac{\mathrm{TP}\times \mathrm{TN}-\mathrm{FP}\times \mathrm{FN}}{\sqrt{\left(\mathrm{TP}+\mathrm{FP})(\mathrm{TP}+\mathrm{FN})(\mathrm{TN}+\mathrm{FP})(\mathrm{TN}+\mathrm{FN}\right)}}$$

### 3.1. Baseline Model Performance Evaluation

Each of the baseline models described in Section 2.3 was assessed using 5-fold cross validation. The multi-modal RGB-D fusion model by Souley Dosso et al. achieved high average sensitivity, specificity, and accuracy, consistently outperforming the exclusively depth-based baseline model across all performance metrics. The cross-validation splits were held constant across models for direct comparison. The rules-based baseline was evaluated by fitting a logistic regression model on the data using the same 5-fold cross-validation splits. The ROC curve of this method can be seen in Figure 9. Table 2 shows a summary of the metrics of the relevant comparison models reproduced from [2].

### 3.2. Comparison Between Baseline Models and Proposed Model

Initially, we compared the results of the depth-based ViT models to those of the baseline depth and RGB-D fusion models. The ‘tiny’ ViT model showed an improvement over all tested metrics except sensitivity, where it showed a slight decrease. The ‘base’ ViT model showed a further improvement over all metrics. Results are summarized in Table 3 and Figure 10. To determine whether the improvement in results observed when moving from the ‘tiny’ model to the ‘base’ model is statistically significant, we performed ANOVA over each of the performance metrics. A *p*-value of less than 0.05 indicated a statistically significant difference in the MCC score when changing the model size.

### 3.3. Effect of Simulated Data on Model Performance

After observing the improved performance of the ViT models on the same data as the baseline models, the tests were repeated with the addition of simulated data into the training folds. These results are shown in Table 4 and Figure 11. Relative to the results in Table 3, the performance decreased with the addition of the simulated data over all metrics except sensitivity. The ANOVA test did not find a statistically significant difference in the results when utilizing the simulated data. This outcome was unexpected, as the addition of the simulated data partially addressed the class imbalance in the training dataset. The decrease in performance could be attributed to domain differences between the simulated and clinical data collection environments or to the difference in class imbalance between the training (7.3:1) and test (10.8:1) datasets. Although efforts were made to recreate the environment when collecting the simulated data, many factors could contribute to the resulting performance, like differences in lighting or mismatch of camera angles between the data collected from the NICU and the simulated data. It is also possible that a larger and more diverse set of simulated data may have a beneficial impact on the models. When looking at the results of the depth-based baseline CNN model, the addition of simulated data showed an increase in specificity and accuracy and a decrease in all other metrics compared to the original depth-based baseline CNN model.

### 3.4. Effect of Perspective Transformation on Model Performance

When repeating the cross validation after applying the perspective transformation process, no pattern of significant increases or decreases in performance could be found (see Table 5 and Figure 12). The ANOVA test did not find any statistically significant effect resulting from pre-processing the images using perspective transformation. The depth-based baseline CNN model showed improvements in most metrics except sensitivity. The difference in the trend of results between the ViT models and the depth-based baseline CNN models may be due to the way each architecture handles images. A CNN uses convolutional operations to learn the patterns of edges and corners in an image, and these features may be enhanced when the perspective of the image is altered. Vision transformers may not benefit in the same way from the enhancement of these features due to the way the ViT splits the input image into patches that are then encoded.

### 3.5. Effect of HHA Encoding on Model Performance

As seen in Figure 13, when comparing the performance of the models using the HHA-encoded depth data against that of the models using the original one-channel depth data, a decrease across all metrics can be seen for the larger-sized ‘base’ vision transformer. However, the smaller ‘tiny’ vision transformer model was shown to improve its specificity, precision, accuracy, and MCC scores, with a stagnant F1-score and a slight decrease in its sensitivity (Table 6). The effect of HHA encoding of the data used to train and evaluate the models was found to be statistically significant for the precision and MCC metrics when applying the ANOVA test. The improvement in the model’s performance was expected, as the model was pre-trained on three-channel RGB images before transferring the weights. Although the depth-based baseline CNN model was pre-trained on the same dataset as the vision transformer models, there was a decrease in the metrics most relevant to the imbalanced dataset being investigated. The depth-based baseline CNN model using HHA-encoded data showed improvements to specificity, accuracy, and precision and a detrimental effect on sensitivity, F1-score, and MCC. These results were surprising since previous studies, such as Gupta et al. [16], have demonstrated that HHA-encoded depth images generally increase the effectiveness of similarly pre-trained CNNs. The deviations observed might stem from the intricate nature of the scenes and the specific conditions of the NICU setting.

### 3.6. Effect of Multiple Variables on Model Performance

The previous four sections outlined the four design parameters applied to the models separately (i.e., model size, simulated data, perspective transform, and HHA encoding). All combinations of the variables were then tested to evaluate their performance and determine the ideal model. This resulted in 11 different combinations of variables (in addition to each variable separately). The results of the remaining models not shown previously can be found in Table 7. An n-way ANOVA was conducted, where *n* = 4 is the number of independent variables. Unexpectedly, it can be seen that no combination of variables was found to have a statistically significant effect on the performance of the models. This may be due to certain variables that have a positive and negative effect counteracting each other when acting in conjunction. In addition, the models may be approaching a performance ceiling as the metrics approach a maximum value that can be achieved with the available input information. Figure 14 and Figure 15 display the metrics for each of the models with and without a combination of variables. The effects of combinations of design variables on the performance of the depth-based baseline CNN model were also investigated. The results of the remaining baseline CNN models not shown previously can be seen in Table 8. The best-performing baseline CNN model utilizes the simulated data as well as HHA encoding. It shows an improvement across all metrics except sensitivity, where the performance decreased. The confusion matrix for the best-performing model can be found in Table 9.

## 4. Discussion

Detecting periods of intervention from a recording presents a number of issues depending on the modality used. RGB video suffers from a decrease in performance during periods of lower light or lighting changes. Models utilizing depth data may be tricked by a nurse’s hands being the same depth away from the camera as the patient or near the patient’s bed. The difference in difficulty of identifying the period of intervention from depth can be seen between Figure 4 and Figure 5. The ‘base’-size vision transformer trained here for the task of intervention detection outperformed the baseline (state-of-the-art) models over all metrics, while the sensitivity of the ‘tiny’ vision transformer was only slightly outperformed by the RGB-D fusion baseline model. When exploring variables that might affect the performance of the models, one of the models trained was a ‘base’ vision transformer that took advantage of HHA encoding of the depth data after applying the perspective transformation process. This model was overall the highest performing model, and its associated confusion matrix can be seen in Table 9. Model size and the encoding type of the depth data were found to have statistically significant effects on the performance of the models, where the ‘base’ model size and HHA encoding were advantageous. Based on the results of our study, we recommend using a vision transformer model with a larger number of trainable parameters applied on depth data that takes advantage of the perspective transformation process outlined in ref. [5] and HHA encoding of depth data, since this approach was found to have the greatest performance in detecting periods of intervention compared to other models tested.

The methods developed here examined individual representative frames from each intervention event, sampled every 30 s, which is in line with the state of the art in RGB-based intervention detection [24]. Our ability to classify the representative frame is expected to reflect the performance of the model when applied to all frames within a continuous period of intervention. We did examine a single period of intervention at greater temporal resolution. For this experiment, we extracted each frame of a 90 s period (2695 frames in total). The period began with no intervention. An intervention (vital sign check and re-swaddling) started after 48 s and continued until the end of the 90 s period. The model (trained on patients different from the test patient) was applied to all 2695 frames, and this performance was compared with the performance estimated from the 14,892 representative frames, originally extracted at 1 frame per 30 s. The resulting performance metrics (Sn = 96.25%, Sp = 100%, Acc = 98.26%, F1 = 98.09%) were equivalent to the performance metrics observed when using the representative frames, validating our approach of evaluating models using representative frames sampled at 1 frame per 30 s. Future work will examine the accuracy with which the precise start and end of each intervention can be determined by the proposed methods. This will be a somewhat nebulous task, since even a human annotator will have difficulty determining the precise start and end points of an intervention (e.g., is the start of the intervention when the clinician’s hands are first visible in frame or when the clinician first makes contact with the patient, etc.).

### Future Work

This paper studied the effect of multiple design parameters (separately and in conjunction) on the performance of ViT clinical intervention detection models. While the use of perspective transform and HHA encoding was found to be beneficial, supplementing the training data with simulated patient care scenes alone did not improve model performance. This outcome was unexpected, as it was thought that correcting for the class imbalance would improve classification accuracy. Although the best-performing model did not include the use of simulated data, the second best model overall did utilize it (ViT Base Simulated HHA PT). This suggests that simulated data may have promise when used in conjunction with other design parameters, and future research could explore the benefits of incorporating more diverse simulated data from a variety of care settings. Researchers could also consider repeating the experiment with simulated data that are captured in an environment that is more comparable to real-world NICU environments. Since the use of HHA encoding had a detrimental effect on some of the baseline depth-based CNN model’s metrics, a possible explanation for the lack of benefit from HHA-encoded data is that the hyperparameter search space may have been insufficient to fine-tune the model and fully leverage these new data. Future work should expand on this search space to re-examine the potential benefit from HHA encoding and consider identifying or training a foundation model pre-trained on HHA-encoded data. Another avenue of research may be to train and test ViT with patches of varying numbers and sizes to study their effect on the performance change that occurs when using perspective transformation. Perspective transformation was shown to increase the performance of the CNN architecture, though the improvement to the ViT model’s performance was not consistent. Future research may also look at background subtraction techniques, where a reference frame containing only the patient is used to highlight differences in depth during an intervention. Lastly, this study examined single-frame depth data; future research will extend this work to consider depth video, since the movements of a clinician in the scene will likely differ from those of the patient. ViT models have recently been extended to RGB video [30,31], and 3D CNN models [32] have also shown great promise for this type of analysis.

## Figures and Tables

**Figure 1 sensors-24-07753-f001:**
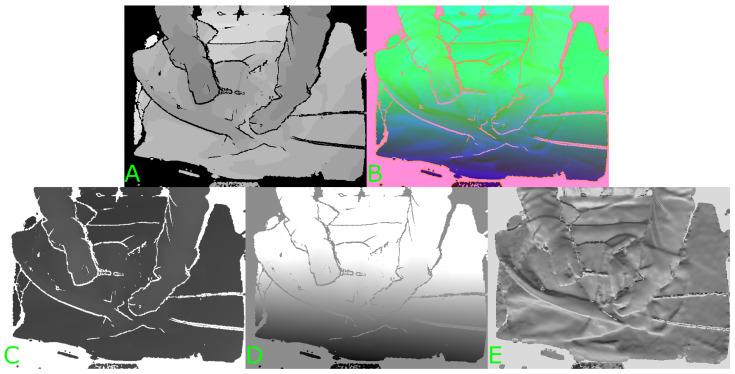
Example of HHA encoding of a depth image. (**A**) Original depth image. (**B**) Three-channel HHA-encoded image. (**C**) First channel (H). (**D**) Second channel (H). (**E**) Third channel (A).

**Figure 2 sensors-24-07753-f002:**
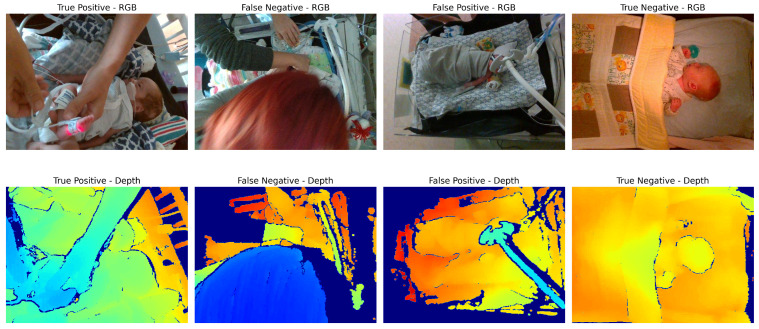
Examples of scenarios where different types of models may underperform. True positive: Image is clear, with distinction between patient and clinician’s hands. False negative: Occlusion of a large portion of the scene by the clinician’s head may confuse depth-based methods (since the clinician’s hands appear to be farther away than the main area of the scene). False positive: Ventilator equipment may be confused for the arm of the clinician performing an intervention. True negative: Patient in crib with no occlusions, angle of camera is top-down which may simplify intervention classification. Note that RGB images are shown here for illustration purposes only; intervention detection models require only depth images.

**Figure 3 sensors-24-07753-f003:**
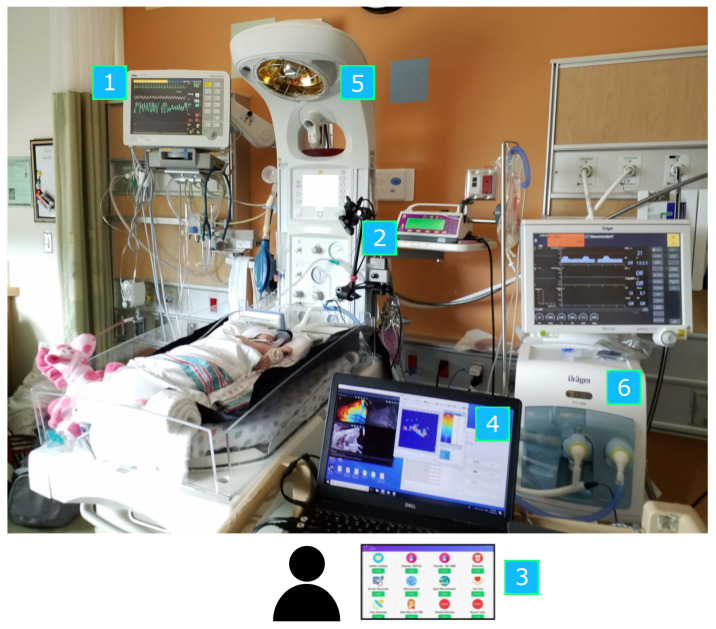
Overview of equipment setup: 1. Patient monitor. 2. RGB-D camera. 3. Bedside annotation application. 4. Data acquisition laptop. 5. Neonatal bed (overhead warmer). 6. Ventilator.

**Figure 4 sensors-24-07753-f004:**
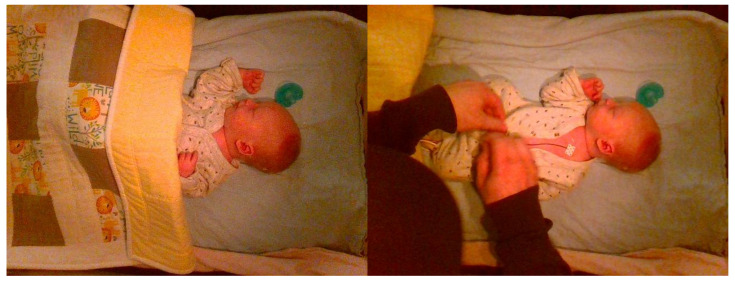
Example frames of ‘No Intervention’ (**left**) and ‘Intervention’ (**right**).

**Figure 5 sensors-24-07753-f005:**
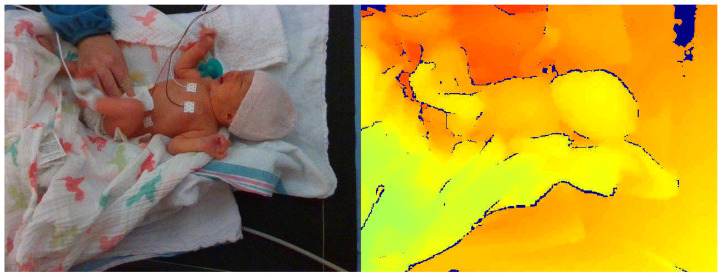
Example of more difficult ‘Intervention’ class frame with both RGB image (**left**) and corresponding depth frame (**right**).

**Figure 6 sensors-24-07753-f006:**
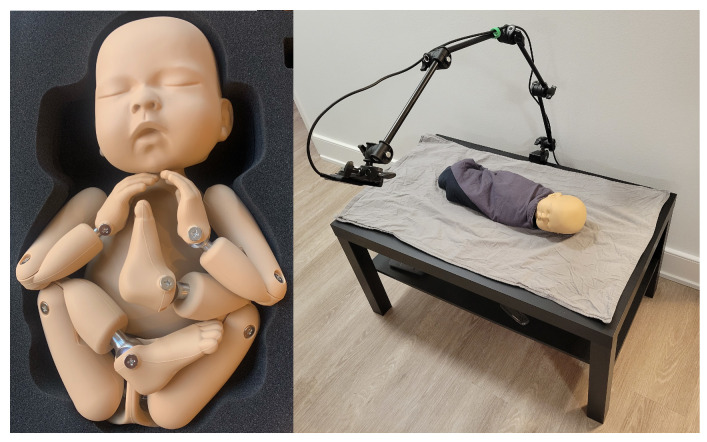
StandInBaby neonatal manikin on the left; example simulated data collection scene on the right.

**Figure 7 sensors-24-07753-f007:**
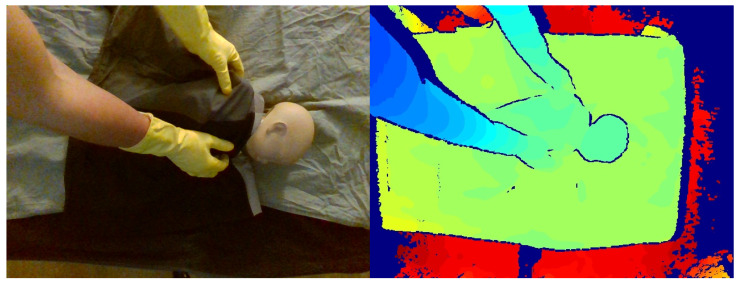
Example of simulated data in RGB (**left**) and colour-mapped depth image (**right**).

**Figure 8 sensors-24-07753-f008:**
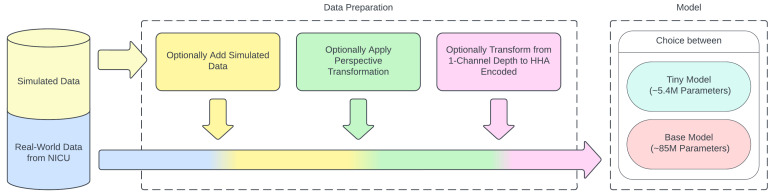
Flow of data and addition of proposed system design parameters.

**Figure 9 sensors-24-07753-f009:**
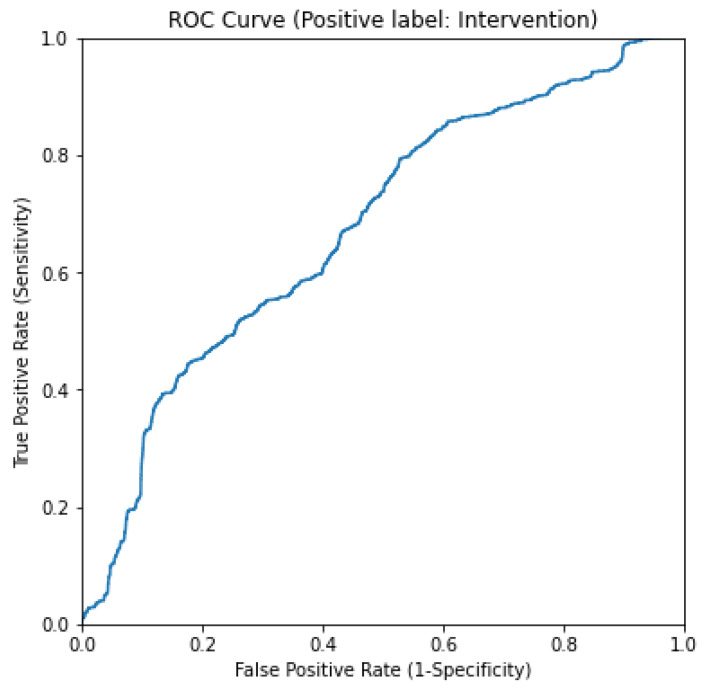
ROC curve for rules-based baseline method.

**Figure 10 sensors-24-07753-f010:**
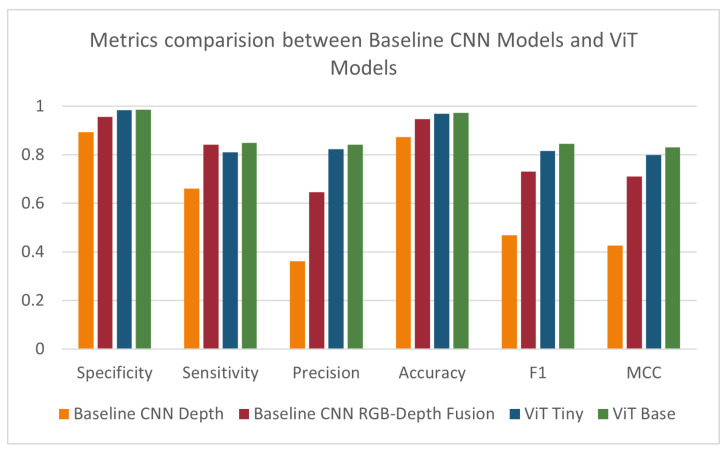
Specificity, sensitivity, precision, accuracy, F1-score, and MCC for baseline models, ViT Tiny, and ViT Base.

**Figure 11 sensors-24-07753-f011:**
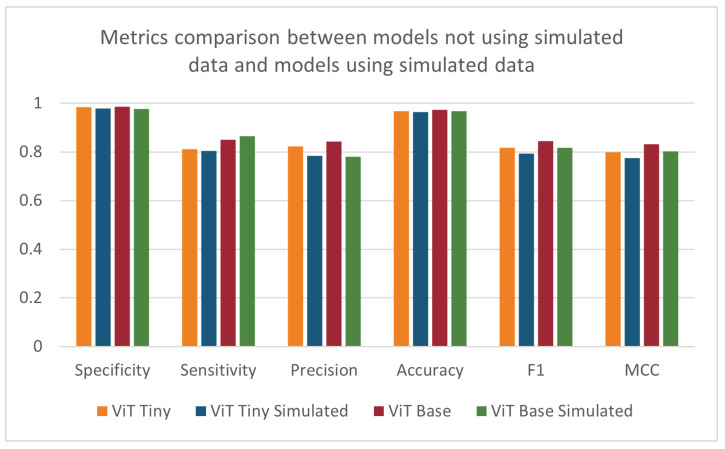
Specificity, sensitivity, precision, accuracy, F1-score, and MCC for ViT models trained with and without supplemental simulated data.

**Figure 12 sensors-24-07753-f012:**
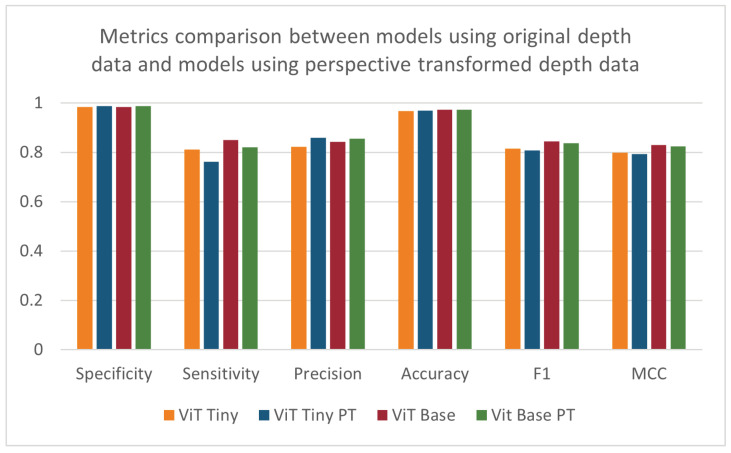
Specificity, sensitivity, precision, accuracy, F1-score, and MCC for models using original depth data and models using perspective-transformed data.

**Figure 13 sensors-24-07753-f013:**
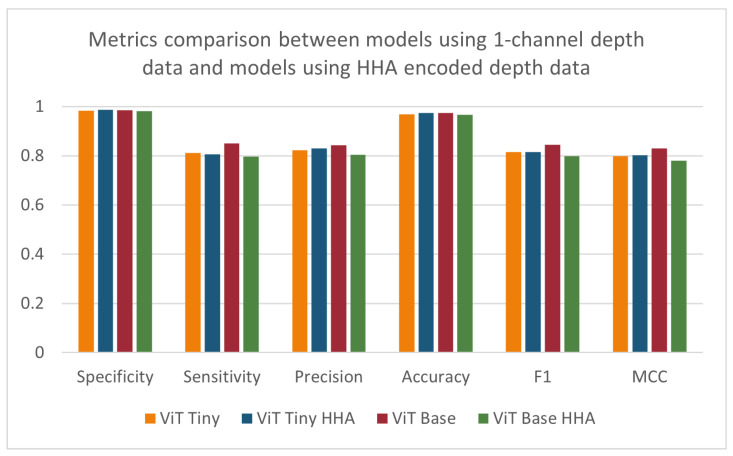
Specificity, sensitivity, precision, accuracy, F1-score, and MCC for models using 1-channel depth data and models using HHA-encoded depth data.

**Figure 14 sensors-24-07753-f014:**
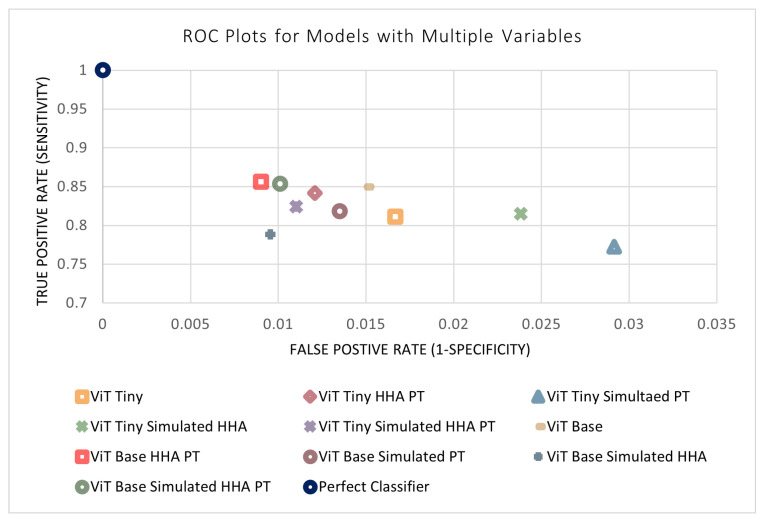
ROC plots of models with a combination of variables.

**Figure 15 sensors-24-07753-f015:**
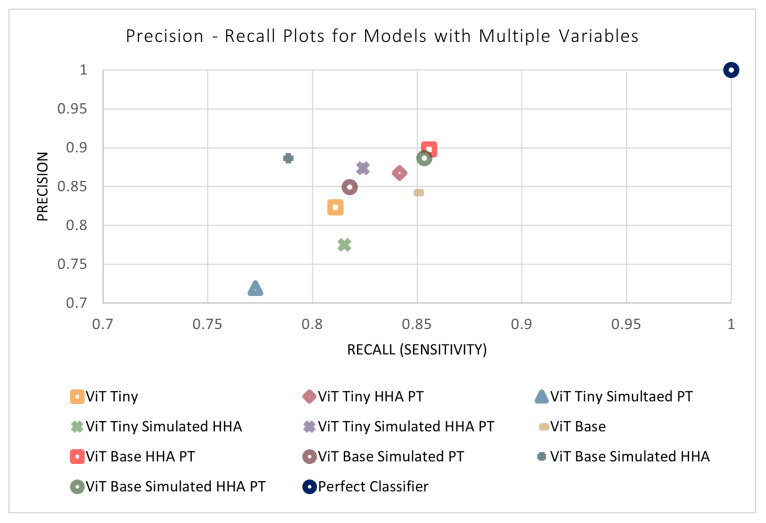
Precision–recall plots of models with a combination of variables.

**Table 1 sensors-24-07753-t001:** Summary of vision transformer experiments.

Experiment	Model Size	Simulated Data	PT	Encoding
1	Tiny	Unused	Unused	1-channel depth
2	Tiny	Unused	Unused	HHA
3	Tiny	Unused	Applied	1-channel depth
4	Tiny	Unused	Applied	HHA
5	Tiny	Added	Unused	1-channel depth
6	Tiny	Added	Unused	HHA
7	Tiny	Added	Applied	1-channel depth
8	Tiny	Added	Applied	HHA
9	Base	Unused	Unused	1-channel depth
10	Base	Unused	Unused	HHA
11	Base	Unused	Applied	1-channel depth
12	Base	Unused	Applied	HHA
13	Base	Added	Unused	1-channel depth
14	Base	Added	Unused	HHA
15	Base	Added	Applied	1-channel depth
16	Base	Added	Applied	HHA

**Table 2 sensors-24-07753-t002:** Summary of results from baseline CNN models used for comparison.

Model	Specificity	Sensitivity	Precision	Accuracy	F1-Score	MCC
RGB-D fusion	**95.70 %**	**84.25%**	**64.54%**	**94.73%**	**73.06%**	**70.98%**
Depth-based	89.25%	66.11%	36.24%	87.29%	46.82%	42.64%
Rules-based	34.97%	43.08%	5.56%	35.63%	9.84%	−12.46%

**Table 3 sensors-24-07753-t003:** Summary of results from ‘tiny’ and ‘base’ vision transformer models.

Model	Specificity	Sensitivity	Precision	Accuracy	F1-Score	MCC
ViT Tiny	98.33%	81.10%	82.29%	96.84%	81.61%	79.93%
ViT Base	**98.50%**	**84.95%**	**84.20%**	**97.35%**	**84.47%**	**83.09%**

**Table 4 sensors-24-07753-t004:** Summary of results from ‘tiny’ vision transformer, ‘base’ vision transformer, and depth-based CNN models with simulated data.

Model	Specificity	Sensitivity	Precision	Accuracy	F1-Score	MCC
ViT Tiny Simulated	97.92%	80.32%	**78.43%**	96.43%	79.28%	77.39%
ViT Base Simulated	97.70%	**86.38%**	77.96%	**96.76%**	**81.78%**	**80.24%**
Depth-based CNN Simulated	**98.07%**	39.19%	65.45%	93.09%	48.94%	47.26%

**Table 5 sensors-24-07753-t005:** Summary of results from ‘tiny’ vision transformer, ‘base’ vision transformer, and depth-based CNN models with perspective-transformed data.

Model	Specificity	Sensitivity	Precision	Accuracy	F1-Score	MCC
ViT Tiny PT	**98.83%**	76.27%	**85.90%**	96.92%	80.72%	79.26%
ViT Base PT	98.72%	**82.16%**	85.66%	**97.32%**	**83.81%**	**82.41%**
Depth-based CNN PT	93.21%	56.48%	44.05%	90.10%	49.17%	44.42%

**Table 6 sensors-24-07753-t006:** Summary of results from ‘tiny’ vision transformer, ‘base’ vision transformer, and depth-based CNN models with HHA-encoded data.

Model	Specificity	Sensitivity	Precision	Accuracy	F1-Score	MCC
ViT Tiny HHA	**98.68%**	80.62%	82.96%	**97.39%**	81.60%	80.30%
ViT Base HHA	98.64%	**83.59%**	**85.08%**	97.37%	**84.25%**	**82.86%**
Depth-based CNN HHA	96.81%	18.95%	40.42%	90.22%	24.36%	22.14%

**Table 7 sensors-24-07753-t007:** Summary of results from ‘tiny’ and ‘base’ vision transformer models with combinations of studied variables.

Model	Specificity	Sensitivity	Precision	Accuracy	F1-Score	MCC
ViT Tiny Simulated PT	97.08%	77.27%	71.91%	95.41%	74.18%	71.92%
ViT Base Simulated PT	98.65%	81.79%	84.93%	97.22%	83.29%	81.82%
ViT Tiny Simulated HHA	97.62%	81.52%	77.50%	96.26%	78.94%	77.24%
ViT Base Simulated HHA	99.04%	78.86%	88.66%	97.34%	83.13%	82.06%
ViT Tiny HHA PT	98.79%	84.16%	86.74%	97.55%	85.38%	84.09%
ViT Base HHA PT	**99.10%**	**85.59%**	**89.76%**	**97.95%**	**87.62%**	**86.54%**
ViT Tiny Simulated HHA PT	98.90%	82.41%	87.39%	97.50%	84.81%	83.51%
ViT Base Simulated HHA PT	98.99%	85.35%	88.64%	97.84%	86.95%	85.80%

**Table 8 sensors-24-07753-t008:** Summary of results from depth-based baseline CNN models with combinations of studied variables.

Model	Specificity	Sensitivity	Precision	Accuracy	F1-Score	MCC
Depth-based CNN Simulated PT	98.83%	36.59%	74.76%	93.56%	48.88%	49.36%
Depth-based CNN Simulated HHA	98.89%	**45.78%**	**79.24%**	**94.39%**	**57.98%**	**57.63%**
Depth-based CNN HHA PT	98.97%	1.90%	20.00%	90.76%	3.32%	3.05%
Depth-based CNN Simulated HHA PT	**99.15%**	33.35%	78.71%	93.50%	46.74%	48.57%

**Table 9 sensors-24-07753-t009:** Confusion matrix for ViT Base HHA PT.

	Predicted	
	**Positive**	**Negative**
**Actual positive**	1083 (TP)	177 (FN)
**Actual negative**	110 (FP)	13,522 (TN)

## Data Availability

The patient data cannot be shared due to Research Ethics Board constraints. The simulated manikin data can be obtained by contacting the authors.

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
