# Peer review of "Depth-Based Intervention Detection in the Neonatal Intensive Care Unit Using Vision Transformers"

_sensors, 2024, doi:10.3390/s24237753_

Round 1
Reviewer 1 Report
Comments and Suggestions for Authors
1) In this manuscript, authors would like to detect the interventions during monitoring the patient in NICU based on videos recorded by the RGB-D camera, which means they should find the different intervention periods from a video (videos). However, I didn't find the direct results that could support the above purpose. The current results were only from binary classification for intervention/non-intervention images at every 30s time point. Generally, Classifying the video clips including different frame images is much more challenging. Or else, the authors should test all the images in a video to check if all the intervention images during the intervention period were correctly detected with the constructed model proposed in this manuscript.
2) Since the imbalance data was used in this work, like 10.8:1 for non-intervention : intervention images from real data and 7.3:1 from real + simulated data, confusion matrix is better used to directly show the classification result for training and test data. Generally, it’s hard to correctly classify the intervention images for such imbalance data even though some other metrics are good like accuracy.
3) For the model training process, authors wrote “The training of the models utilized a mini-batch size of 16 and a learning rate set at 0.01 over a maximum of 15 epochs. ” I wondered if the model has converged after 15 epochs, please show the training history like the training curves of loss or accuracy etc.
4) In the section of Introduction, authors mentioned one contribution of this study is using RGB-D camera other than RGB camera because “RGB video may be considered intrusive and sensitive to ambient lighting.” Thus, it should be more significant to compare the model performance between the RGB-D video and RGB video.
5) The content organization of the manuscript can be further optimized so that the sections of Methods and Results can be more clear. For example, Table 2 is in the section of Methods but it is more suitable in the section of Results.
Author Response
Thank you for your careful review and constructive suggestions. Please see the attached Response to Reviewer file for our detailed response to all reviewer questions and concerns.

Reviewer 2 Report
Comments and Suggestions for Authors
A work for Intervertion Detection in NICUs is proposed focusing on data collection. Some concerns regarding this work:
1. The Introduction is too short and the main contributions of this work should be summarized in the end of introduction instead of the beginning.
2. Introduction and Background should be different sections in order to present the problem in detail.
3. In the proposed method which ViT-based architectures are used for fine tuning? Some details need to be given about the architectures used.
4. More recent methods need to be used for tcomparison of the experiments.
5. The structure of the paper needs to be reviewed.
6. The manuscript needs a careful proofreading.
Comments on the Quality of English LanguageThe manuscript needs a carefull proofreading.
Author Response

(The authors gave the same response as above.)

Round 2
Reviewer 1 Report
Comments and Suggestions for Authors
I would like to thank the authors for answering my questions. However, a number of important clarifications were provided in the responses only and are not reflected in the manuscript. Several responses are also unconvincing. Specifically, I have the following remaining concerns:
1) In this manuscript, the purpose is to automatically detect the interventions during monitoring the patient, which means the constructed model should help find the intervention time period from a video whatever the model was constructed by individual frame images or video clips. The best way to show the work has realized this purpose is to mark the model predicted result (“Intervention”or “No Intervention”) on every frame image of a new video which isn’t used during the model construction so that readers could visually compare the difference between the real situation and the model prediction.
2) From Table 7, I thought the model performance assessment should have some problems. As authors mentioned, 14892 clinical + 600 simulated images in total were used in this manuscript,which means they were already used for model training and validation (“test”word used in the manuscript). Now,14892 clinical images were used again for model test, which hardly demonstrated the model prediction ability since the training, validation and test data was highly correlated and generally resulted in very good model metrics. I worried if the other results on the model performance assessment were obtained by this way. I suggest authors split 14892 images into training and test data with a ratio like 9:1, the training data is used in model construction and test data is only used for model test, and then show model performance on training data and test data separately.
3) I think it’s hard to demonstrate the model has converged from the training curves provided because of the high variation. Authors could show the training curves without the early stopping criterion.
Author Response
Please see the attached response file.
